# The Impact of Tumor Eco-Evolution in Renal Cell Carcinoma Sampling

**DOI:** 10.3390/cancers10120485

**Published:** 2018-12-04

**Authors:** Estíbaliz López-Fernández, José I. López

**Affiliations:** 1FISABIO Foundation, 46020 Valencia, Spain; estibaliz.lopez@yahoo.es; 2Department of Health Sciences, European University of Valencia, Laureate Universities, 46023 Valencia, Spain; 3Department of Pathology, Cruces University Hospital, Biomarkers in Cancer Unit, Biocruces-Bizkaia Health Research Institute, Plaza de Cruces s/n, 48903 Barakaldo, Bizkaia, Spain; 4Department of Medical-Surgical Specialties, University of the Basque Country (UPV/EHU), 48940 Leioa, Spain

**Keywords:** clear cell renal cell carcinoma, tumor evolution, tumor ecology, intratumor heterogeneity, multisite tumor sampling, targeted therapy

## Abstract

Malignant tumors behave dynamically as cell communities governed by ecological principles. Massive sequencing tools are unveiling the true dimension of the heterogeneity of these communities along their evolution in most human neoplasms, clear cell renal cell carcinomas (CCRCC) included. Although initially thought to be purely stochastic processes, very recent genomic analyses have shown that temporal tumor evolution in CCRCC may follow some deterministic pathways that give rise to different clones and sub-clones randomly spatially distributed across the tumor. This fact makes each case unique, unrepeatable and unpredictable. Precise and complete molecular information is crucial for patients with cancer since it may help in establishing a personalized therapy. Intratumor heterogeneity (ITH) detection relies on the correctness of tumor sampling and this is part of the pathologist’s daily work. International protocols for tumor sampling are insufficient today. They were conceived decades ago, when ITH was not an issue, and have remained unchanged until now. Noteworthy, an alternative and more efficient sampling method for detecting ITH has been developed recently. This new method, called multisite tumor sampling (MSTS), is specifically addressed to large tumors that are impossible to be totally sampled, and represent an opportunity to improve ITH detection without extra costs.

## 1. Introduction

Clear cell renal cell carcinoma (CCRCC) is nowadays a health problem of major concern in developed societies. The tumor is an aggressive histologic subtype of renal cancer whose incidence is expected to be increased in the future due to the increasing rate of obesity and the ageing population occurring in Western countries [1]. CCRCC shows a well-known resistance to radio- and chemotherapy [2]. However, anti-angiogenic drugs and immune checkpoint blockade are showing promising therapeutic results. At present, three therapeutic options have demonstrated an overall survival improvement: cabozantinib and nivolumab in second or subsequent lines [3], and the combination of nivolumab and ipilimumab in first line therapy [4].

Pathologists are the medical specialists who handle surgical specimens and decide which parts of the tumor must be included for microscopic, immunohistochemical (IHC) and molecular analyses. Tumor representativeness is becoming a critical issue in modern pathology, either when selecting fragments from large resected tumors or when obtaining tissue cores from tumors prior to surgical resection for diagnostic and/or therapeutic purposes. Increasing evidences are showing in the last times that current sampling strategies may not be giving a complete information about the histological and/or molecular alterations that are present in many tumors of different topographies [5,6,7,8,9]. These inconsistencies rise serious concerns among oncologists [10]. Pathologists, however, seem not to be aware of this central problem since sampling protocols still remain unmodified in routine work.

Intratumor heterogeneity (ITH) is at the basis of the lack of tumor representativeness in many tumor samplings. Although well known for decades, the ITH molecular information is having a significant clinical impact in the last years. The arrival of targeted therapies has increased the need of very precise information about ITH since it is observed in many tumor types and represents a major hurdle for effective therapy, provoking therapeutic resistance and metastatic recurrence [11]. In this sense, Gerlinger et al. [12] unveiled in 2012 to what extent this phenomenon is present in clear cell renal cell carcinomas, describing the molecular heterogeneity of these tumors across different regions within the same tumor.

A new tumor sampling strategy has been developed very recently to improve ITH detection in routine practice [13]. This new method, termed multi-site tumor sampling (MSTS), has been successfully applied to CCRCC, although it can be applied to any tumor large enough as to make impossible a total sampling. Interestingly, MSTS outperforms routine sampling protocols in detecting ITH while keeping the cost fixed [14].

The present paper focuses on the importance of tumor representativeness in modern Oncology. A correct tumor sampling is mandatory for this purpose. CCRCC is a good example since it is a well-known paradigm of ITH. More specifically, this paper revisits some basic concepts and applied clinical issues of CCRCC evolution. Tumor ecology and spatial and temporal evolution constrains are then reviewed to contextualize the urgent need of an appropriate tumor analysis. Also, this overview details a more advantageous alternative for tumor sampling supported by a in silico modeling with clinical validation.

## 2. Tumors as Dynamic Cell Communities Guided by Ecological Principles

Neoplasia encompasses a number of complex and largely unknown processes with a metabolic background [15]. Although pathologists risk thinking of tumors under the microscope as static combinations of cells arranged in different backgrounds organized in varied architectural structures, malignant tumors are complex communities of cells evolving in time with individuals permanently interacting each other. The idea of comparing malignant tumors with other biological communities behaving in a swarm-like manner [16] following similar ecological rules [17,18] is not new. The adaptive mechanisms of tumor cells to their ever-changing habitat are a crucial issue to survive, grow and progress. Epithelial-mesenchymal transition (EMT), and its reversal process (mesenchymal-epithelial transition), are good examples of this dynamic adaptative capacity that can be eventually targeted [19]. A paradoxical effect of EMT is, for example, the development of low-grade metastases in high-grade CCRCC [20]. This tumor behavior, as defined in Ecology, can be synthesized in four models: predation, mutualism, commensalism and parasitism [21]. ITH is known to be induced genetically and epigenetically by interaction with the local tumor microenvironment. Interestingly, tumor microenvironment changes from tumor to tumor, from organ to organ, and even from region to region within the same tumor.

Swarming is defined as the collective behavior of a community of individuals without any centralized guidance or government. This behavior is based essentially on the sum of myriads of neighbor-to-neighbor communications, and is very common in Nature. Tumor cells reproduce this behavior when a specific subgroup within a community of malignant cells of a tumor decides to invade neighbor tissues or metastasize to other organs far away [16]. The collective acquisition of specific properties of tumor cells composing specific tumor compartments is a well-documented event in most tumors, CCRCC included [22]. This phenomenon applies also for tumor microenvironment. For example, a selective loss of PD-L1 expression has been detected in tumor infiltrating lymphocytes (TIL) taking part of the vein/caval tumor thrombi compartment of CCRCC [23].

Predation refers to an inter-individual relationship in which one individual benefits the other killing it. The attack of some T-cells co-localized with tumor cells that is so evident under the microscope in some CCRCC is a good example of predation. This phenomenon has been investigated mainly in breast tumors. For example, a high co-localization of immune and tumor cells was associated to higher 10-year survival in Her2^+^ breast carcinomas [24]. In the same sense, the pattern of the PD-1/PD-L1 axis expression in tumor cells and in TIL predicts tumor aggressiveness and survival also in Her2^+^ breast cancer [25]. Although PD-1/PD-L1 axis blockade is being used with great promise for advanced CCRCC treatment, none study of co-localization of immune and cancer cells in these neoplasms has been published so far.

Cancer cells not always compete for scarce resources. Mutualism applied to cancer refers to the cooperation of two different cell clones for the same benefit thus favoring tumor growth and invasion. This process usually implicates extracellular matrix proteins, such as metalloproteinases and fibronectin, as happens in a zebrafish-melanoma xenograft model [26]. Interestingly, this cooperation has also been identified between tumor cells and stromal inflammatory cells [27].

Examples of commensalism have also been reported in neoplasia. Commensalism describes the relationship of two different cell clones for the benefit of one of them, although the other being not damaged. Also, this process refers to tumor cells and microenvironment cells interactions, for example, the cooperation between tumor cells and tumor-associated fibroblasts favoring tumor progression and metastases [28,29]. High levels of fibroblast activation protein (FAP) have been correlated with tumor size, high grade, high stage and shorter survival in CCRCC, both in primary tumors [30] and in its paired metastases [31]. In this regard, the IHC detection of FAP in the stromal tumor fibroblasts could be a potential biomarker of early lymph node metastatic status and therefore could account for the poor prognosis of FAP positive CCRCC [29]. Even more, recent evidences have shown that tumor microenvironment may vary in an organ-related way along the multiple disseminated metastases of the same tumor [32].

In these collective relationships, some situations lead to an individual to benefit from another damaging it, although not enough so as to destroy it (as happens in predation). In Ecology, this situation is termed parasitism and also occurs in tumor cell communities, for example, when considering the systemic damage generated by the local invasiveness and the metastatic spread of a tumor. This situation could be conceived as a reversal manifestation of the Warburg effect [33], and may explain the frequent association of tumor desmoplasia and biological aggressiveness that occur in many neoplasms.

## 3. Spatial and Temporal Evolution in Clear Cell Renal Cell Carcinomas

As mentioned in previous paragraphs, several researchers have shown how cancer can be regarded from an ecological perspective that governs tumor evolution [17,34,35]. Obvious ethical reasons do not allow the analysis of the temporal evolution of any tumor. However, bioinformatic tools can infer this process by reconstructing the past chronology. Data coming from the molecular analysis of the tumor tissue obtained from the patient can be phylogenetically analyzed. For example, a mathematical modeling has recently defined the timing in the evolution of CCRCC showing that the tumor originates as soon as in the childhood or adolescence of the patient, and remains silent for decades until appearing symptomatic [36]. More exactly, these early events in CCRCC consist in 3p loss with concurrent 5q gain as a result of chromothrypsis, a process that occurs only in a few hundreds of cells [36].

Four models of tumor evolution have been proposed: linear, branched, punctuated and neutral [37]. With respect to the evolutionary patterns, CCRCC may follow branched or punctuated models, as reported very recently [38].

In the linear evolution model, the new sequential driver mutations that appear across the time vanishes the previous ones due to a strong selective advantage. Such tumor evolution was proposed long time ago for some examples of colorectal adenocarcinomas [39]. Linear evolution is an example of Darwinian model.

Branched evolution model has been identified in many human tumors, CCRCC included [12], and basically consists in a truncal mutation shared by all tumor regions followed by clonal, sub-clonal and private mutations across the different tumor regions. The resulting evolutionary trees are shaped by the different accumulative clonal and sub-clonal divergences. Sub-clonal driver mutations and convergent evolution are two features observed in the branched evolution of CCRCC [12,40]. Interestingly, the coexistence of multiple sub-clones in branched tumors opens the door for the possibility of some type of sub-clonal cooperation following any of the ecological patterns described before. Branched evolution also follows a Darwinian model.

Also called the “Big Bang” model [41], the punctuated evolution model is characterized by a large number of genomic changes occurring at very early stages of tumor evolution. ITH is very high at the beginning but decreases progressively across the time as a result of the high selective predominance of very few clones. Typically, tumors developed in a punctuated model display low ITH and high amounts of single chromosomal rearrangements, a phenomenon termed chromothrypsis present in several human tumors, including a subset of CCRCC [36]. Punctuated evolution is also a Darwinian model of tumor evolution.

Neutral evolution is a paradigm of a non-Darwinian model with extremely high ITH in which natural selection driven by sub-clonal mutations and convergent evolution does no take place.

Most CCRCC follow either branched or punctuated evolution models. In this regard, a recent multicenter study analyzing 1206 regions from 101 patients has demonstrated that CCRCC display up to seven distinct evolutionary subtypes [38]. Three of these subtypes (multiple clonal drivers, *BAP1* driven and *VHL* wild type tumors) followed a punctuated model, were associated to aggressive clinical behavior and showed early 9p and 14q losses, high chromosomal complexity and low ITH. By contrast, three other subtypes (*PBRM1* → *SETD2*, *PBRM1* → *PI3K* and *PBRM1* → SCNA driven tumors) followed a branched model and were associated to a less aggressive behavior, with late 9p and 14q losses, low chromosomal complexity and high ITH. The sub-clonal acquisition of *BAP1* or other mutations linked to clinical aggressiveness marked the inflexion from indolence towards rapid evolution in this group of CCRCC. Finally, a seventh subtype (*VHL* mono-driver) was characterized by low chromosomal complexity and low ITH. Typically, this last subtype did not display 9p or 14q losses. These molecular subtypes were associated with classic gross (tumor diameter) and histological (Furhman’s grade, TNM staging, presence of necrosis) parameters and have prognostic implications for patients [38].

With respect to the development of metastases, CCRCC have shown specific routes in a thorough analysis of 575 primary and 335 metastatic biopsies in 100 patients with metastatic CCRCC [42]. This multicenter study included three different cohorts of paired primary and metastatic samples of CCRCC with clinical follow up. The analysis also included samples obtained from the tumor thrombi in 24 cases. In summary, the aggressive evolutionary subtypes, that is, tumors with high chromosomal complexity and low ITH displayed a rapid progression to multiple metastases, whereas the opposite, that is, tumors with low chromosomal complexity but high ITH showed an attenuated temporal tumor progression, with tendency to develop late single metastasis. Finally, tumors within the seventh subtype never metastasized.

Although ITH is a main contributor to the development of therapeutic resistance, the mechanisms that underlie this causal inter-relation provide an example of natural selection through evolutionary adaptation [43]. Genetic and epigenetic changes contribute to modify and adapt tumor cell fitness to the new requirements, thus selecting specific clones not only to invade or metastasize, but also to resist to drugs. Targeted therapies, by definition, select tumor cell populations for resistance, this process being Darwinian in essence [44]. However, therapeutic resistance does not solely concern tumor cells; the local tumor microenvironment also takes part in this process, for example, when the hypoxic status observed in some regions of many tumors applies additional pressure on tumors and contributes to the selection of cell clones adapted to survive under the new conditions [45]. Since drug resistances are a problem of major concern in Oncology and this issue is directly related to regional ITH, a more efficient sampling method to clarify this problem seems mandatory.

## 4. The Need for an Updated Tumor Sampling Adapted to Tumor Type

The only way to discover the complex spectrum of ecological relationships and the spatial and temporal evolution of tumors that have been revisited in previous paragraphs of this narrative is to improve significantly the tumor representativeness with an affordable sampling. In this sense, total tumor sampling would be the ideal solution, but it is not sustainable because many tumors are too large to be analyzed in their entirety. For these cases international accepted protocols were designed. At least theoretically, an effective sampling strategy must assure getting enough tissue so as to provide reliable information in a probable subsequent molecular analysis. The benefit of such strategy should necessarily be balanced with cost in a difficult sustainable equilibrium. At this point the key question is: How extensive must this sampling be to assure tumor representativeness while keeping the costs affordable? or, in other words, when to stop sampling? The answer is complex.

The strategy of getting one tissue fragment (roughly a piece of 1 cm^2^ in dimension) per centimeter of tumor diameter is a rule applied to all tumors, comes from the early days of Pathology and is still followed by pathologists worldwide. When, how and why this strategy was chosen is difficult to know. Although our knowledge about neoplasms has dramatically improved since those days, astonishingly, nothing has changed in tumor sampling. Instead, pathologists focus their interest on the advances provided by -*omics* and other molecular advances brought by sophisticated devices and forget that the most expensive and advanced technique may miss the target if applied in an inappropriate or insufficient tissue sample. Once more, the simplest matters.

Internationally accepted protocols of tumor sampling state that one tumor tissue fragment per centimeter of tumor diameter must be got for histological analysis, plus a fragment of any *suspicious* area [46]. As many tumors appear homogeneous to the naked eye when sliced, tumor selection is usually performed by the pathologist in a blind way and hidden areas of heterogeneity that may be crucial for the patient are usually overlooked.

MSTS has been recently proposed for CCRCC (Figure 1) [13]. This approach follows the rationale *the more you sample the more you find* and applies the *divide and conquer* algorithm [47]. This algorithm has been already applied to resolve complex problems in physics [48], biology [49] and medicine [50], and is based on recursively breaking down a problem in smaller parts (divide) until these are simple enough to be solved directly (conquer). Then, partial solutions are combined to solve the original problem. MSTS has proved to outperform routine protocols in a in silico modeling [14] and in a clinical validation using classic histological parameters [51]. By using MSTS, it is possible to increase the number of samples while keeping the number of cassettes fixed. For such a purpose, the size of the samples must be trimmed from 1 cm^2^ to 3 mm^2^. This way each cassette can contain 8 small samples per cassette (80 in total in a tumor of 10 cm in diameter) obtained from very different, distant and representative regions of the tumor (Figure 2). A straightforward reasoning says that 80 small tissue fragments have a higher chance for detection of ITH as compared to 10 large samples.

If it is accepted that the paraffin block can be the unit of cost in pathology laboratories, it can be assumed that MSTS is better than routine sampling protocol at the same cost [14]. Finally, this method allows the inclusion a small fragment of normal renal tissue in each cassette that can be useful as internal control for immunohistochemistry and/or molecular analyses. The storage of such amount of formalin-fixed and paraffin-embedded tumor tissue fragments in pathology laboratories may be very useful in the future when new technologies allow better analyses. However, pathologists may be reluctant to use MSTS because it takes a long time to collect 80 small samples in a tumor. The application of a cutting grid to the tumor slices in the grossing room will shorten significantly the process [52].

Also, an adaptation of MSTS to large tumors arising in hollow viscera has also been proposed, for example, for cancers arising along the digestive tract. In this particular setting, the anatomical barrier represented by the muscularis propria leads the tumors to grow with their long axes parallel to the wall of the viscera forming tumors with plate-like shapes, and not like spheroids as happens in the kidney, liver or other solid organs. Instead of tissue cubes, the sampling model here gets tissue bars including the whole thickness (from the lumen to the perivisceral adipose tissue) of the viscera [53]. MSTS has obtained here also better performances in detecting ITH compared with routine sampling [53].

As commented in previous paragraphs, temporal evolution in CCRCC follows several deterministic pathways that have been defined either as branched or punctuated models [37]. The discovering of branched CCRCC would benefit specially from MSTS. This possibility makes this strategy even more advantageous in these particular cases. Finally, the performance of MSTS to discover ITH has also proved to be superior at any time of tumor evolution and in all models compared with routine protocols [54].

Recent evidences, however, have demonstrated that tumor sampling must be adapted to the tumor type to be efficient because tumor evolution and the spectrum of ITH are in fact very different, as reflected by the range of evolutionary trees detected across different cancer types [55]. MSTS seems to be an advantageous approach in large tumors with high ITH because in these cases getting samples from many tumor regions will give more complete information of the whole mutational landscape. This happens specially in tumors following branched and neutral models. By contrast, tumors with low ITH levels, although sometimes more aggressive, apparently will not directly benefit from an exhaustive sampling across the tumor. This may happen in neoplasms following the linear and punctuated models.

## 5. Conclusions

This review is addressed to clinicians and pathologists that are involved in Oncology, and revisits the complexity of tumor evolution with a special mention to the reasons for which each tumor is truly unique. The definition of a more efficient method to discover this complexity is an urgent task. Pathologists, the medical specialists who handle surgical specimens, must reconsider if the currently accepted protocols for tumor sampling are appropriate enough to offer with reliability the expected answers that precision medicine needs today. In this regard, new sampling methods could provide substantial advantages for a more precise diagnosis in a subset of cases. MSTS could be a good option since it keeps the balance cost/benefit sustainable.

## Figures and Tables

**Figure 1 cancers-10-00485-f001:**
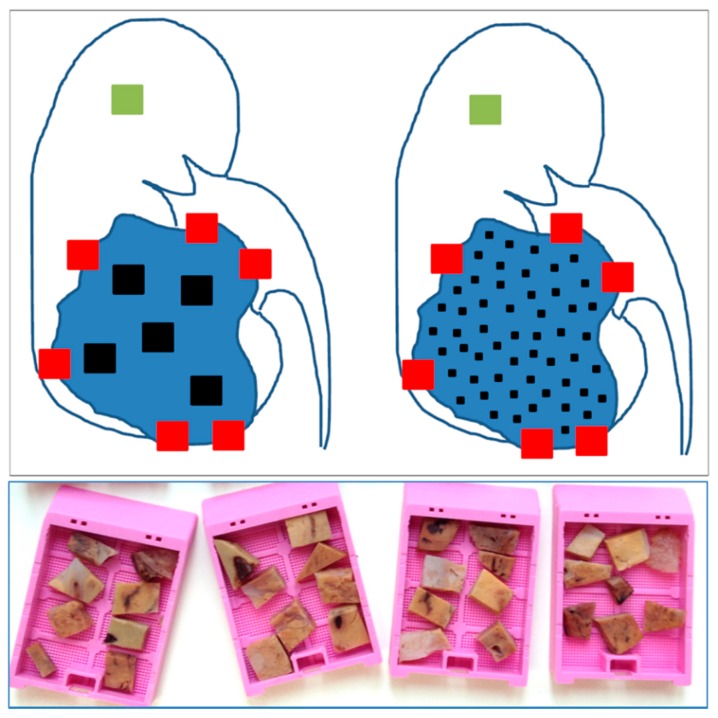
Schematic representation of routine and multisite tumor sampling (MSTS) concepts in renal tumors. The scheme shows the routine sampling (**left**) and the MSTS (**right**). Red cubes represent samples at the edges of the tumor (renal sinus, extrarenal extension and interface between non-tumor and tumor kidney) to detect tumor invasion at these levels. The green cube represents the preceptive sample of normal kidney. Black cubes represent the tumor sampling in both strategies making use of the same number of blocks (up to 8 small fragments can be introduced in one cassette in MSTS). Cassettes containing tumor tissue fragments from MSTS are shown at the bottom of the picture.

**Figure 2 cancers-10-00485-f002:**
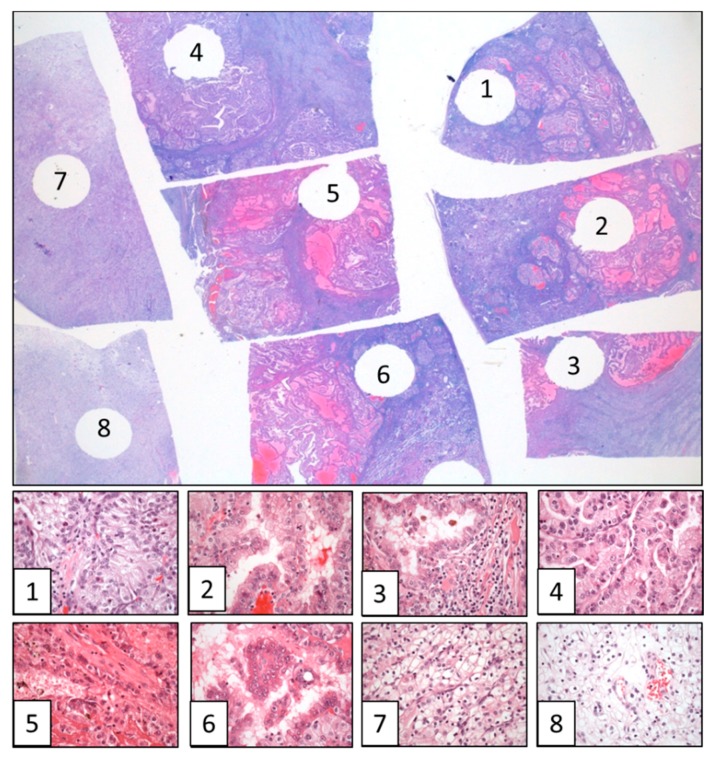
Multisite tumor sampling example in a clear cell renal cell carcinoma (CCRCC) showing the varied spectrum of histologies detected across the eight selected fragments in a single paraffin block, with clear cell high grade phenotype in sample 1, papillary eosinophilic in samples 2, 3, 4 and 6, solid eosinophilic in sample 5, and clear cell low grade in samples 7 and 8. (Hematoxylin & Eosin, original magnification, × 1.5 (large figure) and × 400 (figures 1 to 8)).

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
