# Peer review of "The Impact of Tumor Eco-Evolution in Renal Cell Carcinoma Sampling"

_cancers, 2018, doi:10.3390/cancers10120485_

Round 1
Reviewer 1 Report
Lopez-Fernandez et al., reviewed the impact of MSTS to detect histologically intratumor heterogeneity (ITH) in CCRCCs. The review read well and the content sounds fine, however, there are a few issues to be resolved below.
The ITH is observed in many tumor types and represents a major hurdle for effective therapy, resulting in provoking therapeutic resistance and metastatic recurrence. This concept is supported by the large body of literature (e.g. Marusyk A, et al., Nat Rev Cancer 2012; 12:323-34.).
The ITH is also known to be induced genetically and epigenetically by interaction with tumor microenvironment (TME).
In this review, the authors described the known models of tumor evolution that generate ITH, however there is lack of description about ITH and therapeutic resistance. The reviewer thinks that roles of ITH on therapeutic resistance are good to be discussed.
The authors also indicated that based on genetic alterations CCRCCs with low ITH displayed a rapid progression to multiple metastases and those with high ITH attenuate tumor progression according to ref. 37. Do the authors want to highlight in this review that ITH is associated with less metastatic feature? As mentioned above, The ITH is generated genetically and epigenetically with complexed interaction with TME and it causes therapeutic resistance and metastatic recurrence. The reviewer has a concern that there is the lack of overall description of conceptional roles of ITH on tumor malignancy.
In Fig. 2, the concrete histopathological feature would be described in varied spectrum of histologies (1-8).
Author Response
The ITH is observed in many tumor types and represents a major hurdle for effective therapy, resulting in provoking therapeutic resistance and metastatic recurrence. This concept is supported by the large body of literature (e.g. Marusyk A, et al., Nat Rev Cancer 2012; 12:323-34.).
The sentence and the reference proposed has been included in the Introduction
The ITH is also known to be induced genetically and epigenetically by interaction with tumor microenvironment (TME). Interactions with tumor microenvironment also influence the development of ITH,
The concept has been included in the second section of the paper with some new references and a mention to epithelial-mesenchymal transition process.
In this review, the authors described the known models of tumor evolution that generate ITH, however there is lack of description about ITH and therapeutic resistance. The reviewer thinks that roles of ITH on therapeutic resistance are good to be discussed.
A new paragraph has been included focusing on drug resistance
The authors also indicated that based on genetic alterations CCRCCs with low ITH displayed a rapid progression to multiple metastases and those with high ITH attenuate tumor progression according to ref. 37. Do the authors want to highlight in this review that ITH is associated with less metastatic feature? As mentioned above, The ITH is generated genetically and epigenetically with complexed interaction with TME and it causes therapeutic resistance and metastatic recurrence. The reviewer has a concern that there is the lack of overall description of conceptional roles of ITH on tumor malignancy.
The assertion has been softened in the text
Figure 2 caption
Figure 2 caption briefly describes the histological heterogeneity from 1 to 8
Some spelling errors have also been corrected across the text. All modifications are highlighted in red.
Eight new references have been included
Reviewer 2 Report
The paper entitled “The Impact of Tumor Eco-Evolution in Tumor Sampling” is very interesting and attractive. It deals with an issue of great interest in the anticancer research field that impact the diagnosis of ccRCC: intra-tumor heterogeneity and the subsequent difficulty related to performing an adequate and representative tumor sampling.
I suggest the following minor revisions:
- I suggest to modify the title mentioning the topic of this paper, which is the Renal cell carcinoma.
- Introduction, line 35: only radical surgery has a significant impact in patient’s survival so far. At present, three therapeutic options demonstrated an OS improvement: cabozantinib ad nivolumab in second or subsequent lines, and the combination of nivolumab and ipilimumab in first line therapy. Please mention these studies and re-modulate the sentence above.
Author Response
The title has been modified, as suggested.
Two sentences supported by two new references have been included in the introduction to support drug efficacy in CCRCC treatment.
Some spelling errors have also been corrected across the text. All modifications are highlighted in red.
Eight new references have been included